# Are Antisense Long Non-Coding RNA Related to COVID-19?

**DOI:** 10.3390/biomedicines10112770

**Published:** 2022-11-01

**Authors:** Eman A E Badr, Ibrahim Eltantawy El Sayed, Mohanad Kareem Razak Gabber, Eman Abd Elrehem Ghobashy, Abdullah G. Al-Sehemi, Hamed Algarni, Yasser AS Elghobashy

**Affiliations:** 1Department of Medical Biochemistry and Molecular Biology, Faculty of Medicine, Menoufia University, Shebeen El-Kom 32511, Egypt; 2Department of Chemistry, Faculty of Science, Menoufia University, Shebeen El-Kom 32511, Egypt; 3Faculty of Applied Health Science Technology, Menoufia University, Shebeen El-Kom 32511, Egypt; 4Research Centre for Advanced Materials Science (RCAMS), King Khalid University, P.O. Box 9004, Abha 61413, Saudi Arabia; 5Department of Chemistry, Faculty of Science, King Khalid University, P.O. Box 9004, Abha 61413, Saudi Arabia; 6Department of Physics, Faculty of Science, King Khalid University, P.O. Box 9004, Abha 61413, Saudi Arabia

**Keywords:** LncRNAs, COVID-19, FLVCR1, FLVCR1-DT, A2M-AS1, DBH-AS1

## Abstract

Fighting external pathogens relies on the tight regulation of the gene expression of the immune system. Ferroptosis, which is a distinct form of programmed cell death driven by iron, is involved in the enhancement of follicular helper T cell function during infection. The regulation of RNA is a key step in final gene expression. The present study aimed to identify the expression level of antisense lncRNAs (A2M-AS1, DBH-AS1, FLVCR1-DT, and NCBP2AS2-1) and FLVCR1 in COVID-19 patients and its relation to the severity of the disease. COVID-19 patients as well as age and gender-matched healthy controls were enrolled in this study. The expression level of the antisense lncRNAs was measured by RT-PCR. Results revealed the decreased expression of A2M-AS1 and FLVCR1 in COVID-19 patients. Additionally, they showed the increased expression of DBH-AS1, FLVCR1-DT, and NCBP2AS2. Both FLVCR1-DT and NCBP2AS2 showed a positive correlation with interleukin-6 (IL-6). DBH-AS1 and FLVCR1-DT had a significant association with mortality, complications, and mechanical ventilation. A significant negative correlation was found between A2M-AS1 and NCBP2AS2-1 and between FLVCR1 and FLVCR1-DT. The study confirmed that the expression level of the antisense lncRNAs was deregulated in COVID-19 patients and correlated with the severity of COVID-19, and that it may have possible roles in the pathogenesis of this disease.

## 1. Introduction

The severe acute respiratory syndrome coronavirus 2 (SARS-CoV-2) emerged in December 2019 as the causative agent of the infectious disease COVID-19. SARS-CoV-2 is a positive, single-stranded RNA virus that is structurally similar to SARS-CoV, the causative agent of SARS [1]. 

Most patients with SARS-CoV-2 infection will be asymptomatic or have mild to moderate disease, but severe COVID-19 may develop into pneumonia, pulmonary edema, acute respiratory distress syndrome (ARDS), multiple organ failure, and even death [2,3]. ARDS is considered the leading cause of death from COVID-19, with a mortality rate of around 75%; thus, it is critical to develop a thorough understanding of the pathology, contributory factors, and the possible role of host genetics in variable disease outcomes in order to discover potential treatments [4]. 

Abnormal host response, dysregulated cell death machinery as well as dysregulated gene expression are distinguishing characteristics of several pathologies, including coronavirus disease 2019 [5,6].

As regards abnormal host response, several studies have reported that the death of COVID-19 patients is caused by an extreme release of circulating cytokines such as IL-1, IL-2, IL-6, IL-10, TNF-α, and IFN-γ from their immune system, a condition termed the cytokine storm [7,8,9]. The cytokines’ over secretion is what causes lung damage and even death in COVID-19 patients [10]. 

Ferroptosis is a type of regulated cell death triggered by a combination of iron toxicity, lipid peroxidation, and plasma membrane damage. It may be dysregulated in patients with COVID-19 [11]. Intervention at any level of iron metabolism, such as the degradation of the iron storage protein ferritin, leads to an increase in the concentration of free intracellular iron and may induce ferroptosis [12,13]. 

A heme export protein called Feline Leukemia Virus Subgroup C receptor (FLVCR) facilitates macrophage heme iron recycling [14]. The inhibition of the FLVCR function can lead to the impairment of erythroid maturation and apoptosis. FLVCR-deficient macrophages promote iron overload and the accumulation of ferritin [15,16]. Excess non-protein-bound heme leads to the destruction of lipid, protein, and DNA through the generation of reactive oxygen species, which is followed by cellular injury and death [17].

The ability to avoid infection depends on the adaptive immune system, and during the first critical hours and days of exposure to a new pathogen, we rely upon our innate immune system. Ferroptosis, a distinct form of programmed cell death driven by iron, is involved in regulating CD4+ T cell homeostasis, which enhances the function of follicular helper T cells during infection and following vaccination [18].

Fighting external pathogens relies on the tight regulation of the gene expression of the immune system. The regulation of RNA is a key step in final gene expression, and RNAs can be classified into two categories: RNAs that code for proteins and RNAs that cannot code for proteins, which are referred to as non-coding RNAs (ncRNAs). ncRNAs are subclassified according to their relative size into two types. ncRNAs less than 200 nucleotides are termed small or short non-coding RNAs, while those longer than 200 nucleotides are termed long non-coding RNAs (lncRNAs) [19,20].

LncRNAs regulate many biological processes such as proliferation, differentiation, immune responses [21,22,23,24], and viral pathogenesis [25,26,27,28]. The altered lncRNA expression in COVID-19 has been studied in COVID-19 patients and healthy controls [29,30]. Chen et al. reported that altered lncRNA expression was positively correlated with the severity of COVID-19 patients [31]. Moreover, another study found that the alteration of DANCR and NEAT1 lncRNAs can distinguish between a mild and a severe damage resulting from SARS-CoV-2 [32]. This suggests that the altered expression of lncRNAs may provide a diagnostic and prognostic tool for COVID-19 cases. 

Antisense long non-coding RNAs (antisense lncRNAs) are transcribed from the opposite strand of genes with either a protein-coding or a non-coding function because of nucleotide sequence complementarity, and they have a special role in their corresponding sense gene. Antisense lncRNAs are differentially expressed across different cell types and regulate the expression of specific genes to modulate different signaling pathways [33].

Antisense lncRNAs, including A2M-AS1 (alpha-2-macroglobulin antisense RNA 1), DBH-AS1 (dopamine β hydroxylase antisense RNA 1), FLVCR1-DT (Feline Leukemia Virus Subgroup C Cellular Receptor 1- divergent transcript), NCBP2AS2 (nuclear cap-binding protein subunit-2 antisense RNA 2), and FLVCR1 (Feline Leukemia Virus Subgroup C Cellular Receptor 1) were enrolled in this study of COVID-19 patients to assess their roles in disease severity.

## 2. Materials and Methods

A prospective study was conducted on a total of 180 patients, which included 120 COVID-19 patients as well as 60 age and gender-matched healthy subjects as a control group. The participants were selected from records of confirmed coronavirus-19 patients at the El-Bagour Hospital from March 2021 to December 2021. A written consent form approved by the Local Ethical Research Committee in the Faculty of Medicine of Menoufia University was obtained from the participants before study initiation and after informing them about the study. Every individual in the study was subjected to complete history-taking and a general examination to detect the presence of any systemic disease, and laboratory findings such as CBC (Hb, WBCs count, RBC count, and platelets), CRP, IL6, and D-dimer were also taken into account. PCR nasopharyngeal swab was assessed to confirm the diagnosis of COVID-19, and the gene expression of long non-coding RNA was validated by real-time PCR.

### 2.1. Assessment of COVID-19 Severity

A suggested COVID-19 severity score named the ABCD score was used. This score was based on many parameters, including the age of the patient (<50 years and >50 years), blood tests (leucopenia, lymphocytopenia, CRP level, LDH level, and D-dimer), chest radiograph and CT-scan, comorbidities, and dyspnea. The risk score was developed using these variables, with a minimum score of 0 and a maximum score of 14. The score was subcategorized into 3 groups: mild (0–4), moderate (4–8), and severe (>8). Higher scores indicate increased severity and the need for intensive care. [34]. Chest X-ray, computed tomography, and Reporting and Data System (CO-RADS) were used for image evaluation and for the assessment of the pulmonary involvement of COVID-19 [35]. 

### 2.2. Blood Sample Collection and Preparation

Using sterile venipuncture, six milliliters of fresh venous blood were taken. Two milliliters of blood were immediately transferred to an EDTA vacutainer tube for the total RNA extraction and complete blood count (CBC) via the Sysmex XN-1000 Automated Hematology Analyzer, Kobe, Japan; 2 mL was transferred to a sodium citrate vacutainer tube for the D-dimer test by the Fully Automated Coagulation System, Diagnostic Stago, France. The remaining 2 mL was transferred to a plain tube and centrifuged at 4000 rpm for 20 min. The serum was then refrigerated at −20 °C until the measurements of serum ferritin using the Abbott chemiluminescence immunoassay, North Chicago, IL, USA; of the C-reactive protein (CRP) using the nephelometric method by Mispa-i2, and of interleukin 6 (IL-6) using the enzyme-linked immunosorbent assay (ELISA) human kits.

### 2.3. Measurement of Long Non-Coding RNA Expression by Real-Time PCR Technique

The total RNA was extracted using Thermo-Scientific’s Gene-JET Blood RNA Purification Kit. The concentrations and purity of the RNA extract were determined by Nano DropTM 2000, Thermo Scientific, Waltham, MA, USA.

Until the reverse transcription process, the RNA extract was stored frozen at −80 °C, employing the master mix of cDNA synthesis, Thermo Scientific’s RevertAid First Strand cDNA Synthesis Kit prepared on ice, with a total reaction volume of 20 μL: First, 10 μL of RNA was mixed with 1 μL of hexamer primer and 1 μL of nuclease-free water, which was incubated at 65 °C for 5 min and then chilled on ice. Second, 4 μL of 5 X reaction buffer, 1 μL of RNase inhibitor, 2 μL of 10 mM dNTPs, and 1 μL of Revertaid RT were added to the above-mentioned mix to make a total volume of 20 μL. The samples were incubated for a single cycle of incubation, as follows: 5 min at 25 °C, 60 min at 42 °C, and 5 min at 70 °C on the 2720 thermal cycler (ABI systems, Singapore). 

In the real-time PCR stage, 10 μL of Sensi FAST TM SYBR green Lo-ROX, 1 μL of nuclease-free water, 6 μL of cDNA, and 1.5 μL of each forward and reverse primers (listed in the Appendix A) were mixed to obtain a total reaction volume of 20 μL for each sample. Using the 7500 (v.2.0.1; Applied Biosystems, Waltham, MA, USA), the samples were incubated at 95 °C for 10 min, followed by 50 cycles at 95 °C for 15 s, 60 °C for 1 min, 72 °C for 1 min, and finally at 72 °C for 10 min. The relative expression of the long non-coding RNA was determined using the 2^−ΔΔCt^ technique, which was normalized to the endogenous housekeeping gene (beta-actin) and compared to the control, with Δ Ct = Ct target − Ct reference, ΔΔ Ct = (Δ Ct sample − Δ Ct control). 

### 2.4. Statistical Analysis of the Data

The collected data were tabulated and analyzed with an IBM personal computer using the SPSS software package version 20.0. (Armonk, NY, USA: IBM Corp). The distribution of variables was verified via the nonparametric test, Kolmogorov–Smirnov. A comparison of two categorical variables was assessed by the Chi-square test, while a comparison of two groups of not normally distributed quantitative variables, the Mann–Whitney test was used. The Kruskal–Wallis test was used for the comparison of two or more groups of not normally distributed quantitative variables, followed by a post hoc test (Dunn’s for multiple comparisons test) for pairwise comparison. A one-way analysis of variance test (ANOVA) was used for the comparison of more than two groups, followed by the post hoc test (Tukey) for pairwise comparison. The receiver operating characteristic curve (ROC) was used to determine the diagnostic performance of the markers. The Spearman coefficient was used to correlate between quantitative variables. Univariate and multivariate regression analyses were performed to calculate the effects of risk factors as independent. Values were considered significant when less than 0.05.

## 3. Results

Controls and two patient groups were age and gender-matched. The mean age of the control subjects was 58.77 ± 13.41 years, that for patients with moderate COVID-19 infection was 54.38 ± 16.21 years, and that for patients with severe COVID-19 infection was 59.22 ± 11.65 years (*p* = 0.112). Males represented 58.3% of the control group, 55.0% of patients with moderate COVID-19 infection, and 56.7% of patients with severe COVID-19 infection (*p* = 0.934). Regarding the vital signs, the systolic and diastolic blood pressure did not show significant differences among the different groups (*p* = 0.080 and 0.179). The mean respiratory rate was significantly higher and the mean oxygen saturation was significantly lower among patients with severe COVID-19 infection (*p* < 0.001 for both) (Table not shown).

The WBC and platelet counts were significantly lower, while the serum ferritin, CRP, IL6, and D-dimer were significantly higher among patients with moderate and severe COVID-19 infection as compared to controls (*p* ≤ 0.001 for all). Furthermore, patients with severe COVID-19 infection had significantly higher levels of serum ferritin, CRP, and IL6 than patients with moderate COVID-19 infection (Table 1).

In a comparison of the long non-coding RNAs between the studied groups, the A2M-AS1 and FLVCR1 showed a non-significant difference between the controls and patients with moderate COVID-19 infection (*p* = 0.368 and 0.092), while patients with severe COVID-19 infection had a significantly lower level of both as compared to controls and patients with moderate COVID-19 infection (*p* =< 0.001). The DBH-AS1, FLVCR1-DT, and NCBP2AS2-1 RNAs showed significantly higher levels in patients with moderate COVID-19 infection than controls, while patients with severe infection had significantly higher levels than both controls and patients with moderate infection (*p* =< 0.001 for all). To assess the pulmonary involvement of COVID-19 among patient groups, the Reporting and Data System (CO-RADS) was used, which showed a higher suspicion in patients with severe COVID-19 infection [3] than in patients with moderate infection [2] (Table 2).

In studying the percentage of complications, mechanical ventilation, and mortality among patients with severe COVID-19 infection, the studied group had the following results: 25%, 15%, and 10%, respectively (Table 3).

The Spearman correlation between different clinical and laboratory markers with long non-coding RNAs in patients with severe COVID-19 infection showed a significant negative correlation between A2M-AS1, FLVCR1, and ferritin levels, while there was a significant positive correlation between the FLVCR1-DT and ferritin levels. FLVCR1-DT also had a significant positive correlation with IL6 (*p* =< 0.001), while NCBP2AS2-1 had a significant negative correlation with lymphocyte count and a significant positive correlation with IL6 (*p* =< 0.001) (Table 4).

In studying the correlation between the types of long non-coding RNAs in patients with severe COVID-19 infection, a significant positive correlation was found between A2M-AS1 and FLVCR1 as well as between DBH-AS1 and both FLVCR1-DT and NCBP2AS2-1. A significant negative correlation was found between A2M-AS1 and NCBP2AS2-1 and between FLVCR1 and FLVCR1-DT (Table 5).

The Receiver Operating Characteristic (ROC) curve examination of different long non-coding RNAs in the study to discriminate between severe and moderate COVID-19 infection showed that: For A2M-AS1, the best cutoff was ≤0.98, with a sensitivity, specificity, PPV, and NPV of 98.33%, 75.0%, 79.7%, and 97.8%, respectively. For FLVCR1, the best cutoff was ≤0.82, with a sensitivity, specificity, PPV, and NPV of 63.33%, 61.67%, 62.3%, and 62.7%, respectively. For DBH-AS1, the best cutoff was >9, with a sensitivity, specificity, PPV, and NPV of 75.0%, 81.67%, 80.4%, and 76.6%, respectively. For FLVCR1-DT, the best cutoff was >8.6, with a sensitivity, specificity, PPV, and NPV of 71.67%, 70.0%, 70.5%, and 71.2%, respectively. For NCBP2AS2-1, the best cutoff was >3.55, with a sensitivity, specificity, PPV, and NPV of 91.67%, 85.0%, 85.9%, and 91.1%, respectively (Table 6).

To adjust the confounding factors affecting mortality, complications, and mechanical ventilation that are associated with severe COVID-19 infection, a univariate logistic regression analysis was performed, and the results showed that A2M-AS1, DBH-AS1, and FLVCR1-DT had a significant association with mortality, while DBH-AS1, FLVCR1-DT, and NCBP2AS2-1 had a significant association with complications and mechanical ventilation. A multivariate logistic regression analysis showed that only FLVCR1-DT had been associated with complications, while FLVCR1-DT and NCBP2AS2-1 are associated with mechanical ventilation (Appendix A).

## 4. Discussion

COVID-19, caused by the new coronavirus SARS-CoV-2, has an exaggerated inflammatory response that can lead to severe manifestations such as acute respiratory distress syndrome, sepsis, coagulopathy, and death in a proportion of patients [36].

The course of COVID-19 includes the inflammatory cytokine storm, acute respiratory distress, and multiorgan failure with elevated levels of IL-6, D-dimer, serum ferritin, and C-reactive protein (CRP) [37,38,39]. Cytokine storms might be caused by activated lymphocytes and macrophages induced by viral infections [40].

These agree with the findings of the present study, which showed a significant increase in IL6, D-dimer, CRP, and serum ferritin in both patient groups as compared to controls and in patients with severe COVID-19 as compared to moderate cases.

Inflammatory cytokines can directly damage the alveolar structure and pulmonary vasculature, thus promoting alveolar edema and leading to the dysfunction of pulmonary ventilation [41,42,43,44,45]. Several studies have reported that the level of inflammatory cytokines, especially IL-6, is correlated with COVID-19 severity and could also predict the need for mechanical ventilation [46,47,48].

Deregulated lncRNAs in SARS-CoV-2 infection are associated with many biological processes and pathways, such as the MAPK signaling pathway, innate immune response cytokine–cytokine receptor interaction, oxidative stress, viral defense response, innate immune response, and inflammatory response [49]. However, the detailed biological mechanism of these lncRNAs in the pathogenesis of the COVID-19 disease is still under study.

The lncRNA A2M antisense RNA1 (A2M-AS1) is a novel lncRNA identified as one of the upstream regulators that might affect the expression of ferroptosis-related genes [18].

The α-2-macroglobulin (A2M) acts as a carrier of pro-inflammatory cytokines such as interleukin-6 and tumor necrosis factor-α, while A2M-AS1 has proliferation and anti-apoptosis effects that silence A2M-AS1 and lead to anti-proliferation and pro-apoptosis effects [50].

High intracellular iron levels are associated with the non-apoptotic cell death pathway called ferroptosis. The Feline Leukemia Virus Subgroup C has been identified as having roles in cellular heme transport and in the control of intracellular heme content as heme synthesis increases to support erythrocyte differentiation [51]. Free heme may be exported by the Feline Leukemia Virus Subgroup C receptor 1 or 2 (FLVCR1/2) or converted into inorganic Fe by heme oxygenase [52].

Feline Leukemia Virus Subgroup C cellular receptor 1 (FLVCR1) is a heme exporter; FLVCR1a resides in the plasma membrane and is responsible for heme detoxification, while FLVCR1b is located in mitochondria and is involved in the transport of newly synthesized heme from the mitochondria to the cytosol [53]. The expression of FLVCR1 is needed to control the size of the cytoplasmic free heme pool, which is essential for proper metabolic functions [54].

The expression levels of both A2M-AS1 and FLVCR1 in the present study showed a significant decrease in patients with severe COVID-19 as compared to those in controls and patients with moderate COVID-19.

Dopamine β hydroxylase antisense RNA 1(DBH-AS1) is a ~2kb lncRNA transcribed from chromosome 9q34, with a polyadenylated tail transcribed from chromosome 9q34 [55]. Dendritic cells (DCs) are antigen-presenting cells, and the abnormal function of these cells causes immune-related diseases. DBH-AS1 can modulate several cell-cycle-related factors, leading to enhanced proliferation and resistance to apoptosis [56].

Nuclear cap-binding protein subunit-2 antisense RNA 2 (NCBP2AS2), also called the hypoxia-induced angiogenesis regulator (HIAR), is abundant in hypoxic cancer-associated fibroblasts (CAFs). The silencing of HIAR abolishes the pro-angiogenic and pro-migratory function of hypoxic CAFs by decreasing the secretion of the pro-angiogenic factor in endothelial cells [57].

The cap-binding complex (CBC), which is directly associated with the RNA cap and serves as an adaptor for other RNA processing factors, is comprised of the nuclear cap-binding protein 2 (NCBP2) and NCBP1. NCBP2 and NCBP3 function redundantly to support the nuclear processing of mRNAs. The antiserum to NCBP2 or NCBP3 independently co-precipitate NCBP1. NCBP3-deficient cells evoke a reduced antiviral response [58,59].

The expression levels of DBH-AS1, FLVCR1-DT, the antisense long non-coding RNA of FLVCR1 and NCBP2AS2-1, and the antisense long non-coding RNA of NCBP2 in COVID-19 patients in this study were deregulated, with a significant increase in both patient groups in comparison with controls.

An examination of the ROC curve of these lncRNAs revealed that A2M-AS1 and NCBP2AS2-1 had the best validity to discriminate between severe and moderate COVID-19 infection, and measuring the expression levels of these lncRNAs in COVID-19 patients could be useful in prognosis and in monitoring the treatment of these patients.

The present study revealed a significant decrease in platelet count in both patient groups as compared to controls, and this agrees with large-scale studies that have reported the presence of thrombocytopenia in about 18.8–36.2% of COVID-19 patients [60,61].

Another cause of thrombosis in COVID-19 patients is the critical role of activated platelets, which adhere to the sub-endothelium, leading to pulmonary embolism and arterial ischemia [62,63].

Furthermore, the results revealed that FLVCR1-DT is associated with complications, while FLVCR1-DT and NCBP2AS2-1 are associated with mechanical ventilation as an indicator of the severity of the disease. This could be due to a change in free iron content by FLVCR1-DT, leading to a change in the ferroptosis process and an increase in the secretion of the pro-angiogenic factor in the endothelial cells by NCBP2AS2-1.

The results revealed a significant positive correlation of IL6 with both FLVCR1-DT and NCBP2AS2-1. In agreement with our results, a previous study reported that the expression of IL-6 is regulated by lncRNAs through NFκB, JAK/STAT, MAPK, and NFκB [64], and these findings support the synergistic role of the cytokine storm and activated platelets with deregulated lncRNAs in the pathogenesis of the COVID-19 disease.

## 5. Conclusions

This study confirmed that the gene expression profiling of the four antisense lncRNAs in COVID-19 patients was deregulated and altered and may have possible roles in the pathogenesis of this disease, together with their roles in discriminating severe from moderate cases. More studies should be carried out in order to examine the expression levels of these lncRNAs in treated COVID-19 patients.

## Figures and Tables

**Table 1 biomedicines-10-02770-t001:** Comparison between the three studied groups according to the lab investigation.

Lab Investigation	Group I (*n* = 60)	Group II (*n* = 60)	Group III (*n* = 60)	Test of Significance	*p*
**Hemoglobin (gm/dL)**					
Mean ± SD	12.32 ± 1.41	12.41 ± 1.50	11.88 ± 1.55	F = 2.126	0.122

**WBC (×103/uL)**					
Mean ± SD	7.11 ± 1.47	2.74 ± 0.36	2.73 ± 0.28	F = 488.252 *	<0.001 *

**Significance between Groups**	*p*_1_ < 0.001 *_,_ *p*_2_< 0.001 *, *p*_3_ = 0.998				
**Platelets**					
Median (Min–Max)	272.0 (178.0–397.0)	127.0 (115.0–172.0)	130.0 (115.0–138.0)	H = 123.172 *	<0.001 *

**Significance between Groups**	*p*_1_ < 0.001 *_,_ *p*_2_ < 0.001 *, *p*_3_ = 0.067				
**Lymphocytes (10^3^/uL)**					
Median (Min–Max)	1.60 (1.30–1.90)	1.60 (1.20–1.90)	1.50 (1.30–16.0)	H = 3.550	0.169

**D-dimer (mg/mL)**					
Median (Min–Max)	0.18 (0.05–0.30)	1.40 (1.10–2.50)	1.35 (1.10–1.60)	H = 123.486 *	<0.001 *

**Significance between Groups**	*p*_1_ < 0.001 *_,_ *p*_2_ < 0.001 *, *p*_3_ = 0.107				
**CRP (mg/L)**					
Median (Min–Max)	1.90 (1.10–7.20)	72.0 (24.0–96.0)	96.0 (96.0–192.0)	H = 145.989 *	<0.001 *

**Significance between Groups**	*p*_1_ < 0.001 *, *p*_2_ < 0.001 *, *p*_3_ < 0.001 *				
**Ferritin (ng/mL)**					
Mean ± SD	30.33 ± 3.17	35.03 ± 3.72	96.22 ± 8.58	F = 2496.411 *	<0.001 *

**Significance between Groups**	*p*_1_ < 0.001 *, *p*_2_ < 0.001 *, *p*_3_ < 0.001 *				
**IL6 (pg/mL)**					
Mean ± SD	1.94 ± 0.60	7.77 ± 1.70	11.62 ± 1.23	F = 900.400 *	<0.001 *

**Significance between Groups**	*p*_1_ < 0.001 *, *p*_2_ < 0.001 *, *p*_3_ < 0.001 *				

SD: Standard deviation; F: F for ANOVA test, pairwise comparison between two groups were performed using the post hoc test (Tukey); H: H for Kruskal–Wallis test, pairwise comparison between two groups were performed the using post hoc test (Dunn’s for multiple comparisons test); *p*: *p*-value for comparison of the studied groups; *p*_1_: *p*-value for comparison of Group I and Group II; *p*_2_: *p*-value for comparison of Group I and Group III; *p*_3_: *p*-value for comparison of Group II and Group III. *: Statistically significant at *p* ≤ 0.05. Group I: Control group; Group II: Moderate COVID-19 infection; Group III: Severe COVID-19 infection.

**Table 2 biomedicines-10-02770-t002:** Comparison between the three studied groups according to different parameters.

	Group I (*n* = 60)	Group II (*n* = 60)	Group III (*n* = 60)	Test of Significance	*p*
**A2M-AS1**					
Median (Min–Max)	1.75 (0.58–2.80)	1.65 (0.20–4.78)	0.44 (0.02–1.0)	H = 100.629 *	<0.001 *

**Significance between Groups**	*p*_1_ = 0.368, *p*_2_ < 0.001 *, *p*_3_ < 0.001 *				
**FLVCR1**					
Median (Min–Max)	1.38 (0.08–3.83)	0.94 (0.12–4.56)	0.48 (0.01–11.25)	H = 24.269 *	<0.001 *

**Significance between Groups**	*p*_1_ = 0.092, *p*_2_ < 0.001 *, *p*_3_ = 0.002 *				
**DBH-AS1**					
Median (Min–Max)	2.45 (0.01–7.62)	5.85 (0.19–11.0)	14.58 (0.25–51.0)	H = 68.041 *	<0.001 *

**Significance between Groups**	*p*_1_ < 0.001 *, *p*_2_ < 0.001 *, *p*_3_ < 0.001 *				
**FLVCR1-DT**					
Median (Min–Max)	0.70 (0.30–1.0)	7.45 (2.40–11.0)	12.70 (2.0–77.0)	H = 131.159 *	<0.001 *

**Significance between Groups**	*p*_1_ < 0.001 *, *p*_2_ < 0.001 *, *p*_3_ = 0.001 *				
**NCBP2AS2-1**					
Median (Min–Max)	0.45 (0.13–1.10)	2.81 (1.51–3.81)	8.98 (2.13–45.77)	H = 151.736 *	<0.001 *

**Significance between Groups**	*p*_1_ < 0.001 *, *p*_2_ < 0.001 *, *p*_3_ < 0.001 *				
**CO-RADS**					
Median (Min–Max)	–	2.0 (1.0–2.0)	3.0 (3.0–5.0)	U = 0.0 *	<0.001 *


SD: Standard deviation; F: F for ANOVA test, pairwise comparison between two groups were performed using the post hoc test (Tukey); H: H for the Kruskal–Wallis test, pairwise comparison between two groups were performed using the post hoc test (Dunn’s for multiple comparisons test); *p*: *p*-value for comparing the studied groups; *p*_1_: *p*-value for the comparison of Group I and Group II; *p*_2_: *p*-value for the comparison of Group I and Group III; *p*_3_: *p*-value for the comparison of Group II and Group III. *: Statistically significant at *p* ≤ 0.05. Group I: Control group; Group II: Moderate COVID-19 infection; Group III: Severe COVID-19 infection.

**Table 3 biomedicines-10-02770-t003:** Distribution of the studied cases according to complications, mechanical ventilation, and mortality in group III (*n* = 60).

	No. (%)
**Complications**	
No	45 (75.0%)
Yes	15 (25.0%)
**Mechanical ventilation**	
No	51 (85.0%)
Yes	9 (15.0%)
**Mortality**	
No	54 (90.0%)
Yes	6 (10.0%)

**Table 4 biomedicines-10-02770-t004:** Correlation between different parameters in group III (*n*= 60).

	A2M-AS1		FLVCR1		DBH-AS1		FLVCR1-DT		NCBP2AS2-1	
	r_s_	*p*	r_s_	*p*	r_s_	*p*	r_s_	*p*	r_s_	*p*
**Age (years)**	0.006	0.962	−0.112	0.392	**0.355**	**0.005 ***	0.009	0.945	0.017	0.898
**SBP**	0.086	0.514	0.252	0.052	−0.142	0.279	−0.098	0.459	−0.154	0.241
**DBP**	−0.067	0.609	0.157	0.230	−0.189	0.148	−0.095	0.422	−0.074	0.576
**RR**	−0.132	0.315	0.046	0.725	−0.050	0.704	−0.075	0.568	0.034	0.799
**Temp**	0.034	0.794	0.095	0.468	−0.087	0.507	−0.073	0.529	−0.007	0.960
**HR**	−0.069	0.598	−0.072	0.584	0.039	0.769	0.009	0.948	−0.133	0.309
**Oxygen saturation**	0.028	0.834	0.142	0.281	0.033	0.802	−0.098	0.458	−0.036	0.787
**Hemoglobin (gm/dL)**	−0.063	0.634	0.059	0.729	0.039	0.769	0.043	0.833	0.141	0.283
**WBC (×10^3^/uL)**	0.039	0.766	−0.063	0.634	−0.269	0.038 *	−0.023	0.864	0.096	0.467
**Platelets**	0.023	0.863	−0.014	0.914	0.083	0.528	0.042	0.750	0.197	0.132
**Lymphocytes (10^3^/uL)**	−0.118	0.370	0.136	0.300	−0.098	0.458	0.125	0.342	**−0.268**	**0.038 ***
**D-dimer (mg/mL)**	0.084	0.523	−0.011	0.935	−0.038	0.775	0.047	0.724	0.044	0.738
**CRP (mg/L)**	0.068	0.603	0.112	0.393	−0.082	0.535	0.071	0.587	−0.115	0.380
**Ferritin (ng/mL)**	**−0.355**	**0.005 ***	**−0.294**	**0.021 ***	−0.065	0.620	**0.295**	**0.022 ***	0.116	0.379
**IL6 (pg/mL)**	0.164	0.210	−0.162	0.216	−0.074	0.572	**0.868**	**<0.001 ***	**0.848**	**<0.001 ***

rs: Spearman coefficient. *: Statistically significant at *p* ≤ 0.05.

**Table 5 biomedicines-10-02770-t005:** Correlation between different parameters in group III.

		A2M-AS1	FLVCR1	DBH-AS1	FLVCR1-DT	NCBP2AS2-1
**A2M-AS1**	**r_s_**	1.000	0.282	−0.089	−0.083	−0.396
	** *p* **		0.029 *	0.497	0.526	0.002 *
**FLVCR1**	**r_s_**			−0.069	−0.339	−0.211
	** *p* **			0.601	0.009 *	0.105
**DBH-AS1**	**r_s_**				0.304	0.311
	** *p* **				0.018 *	0.016 *
**FLVCR1-DT**	**r_s_**					0.238
	** *p* **					0.067
**NCBP2AS2-1**	**r_s_**					
	** *p* **					

rs: Spearman coefficient. *: Statistically significant at *p* ≤ 0.05.

**Table 6 biomedicines-10-02770-t006:** Validity (AUC, sensitivity, specificity) for different parameters to discriminate severe group (*n* = 60) from moderate group (*n* = 60).

	AUC	*p*	95% C.I	Cutoff	Sensitivity	Specificity	PPV	NPV
**A2M-AS1**	0.934	<0.001 *	0.892–0.976	≤0.98	98.33	75.0	79.7	97.8
**FLVCR1**	0.653	0.001 *	0.585–0.780	≤0.82	63.33	61.67	62.3	62.7
**DBH-AS1**	0.802	<0.001 *	0.715–0.890	>9	75.0	81.67	80.4	76.6
**FLVCR1-DT**	0.771	<0.001 *	0.683–0.860	>8.6	71.67	70.0	70.5	71.2
**NCBP2AS2-1**	0.951	<0.001 *	0.902–1.0	>3.55	91.67	85.0	85.9	91.1

AUC: Area under the curve. *p* value: Probability value. CI: Confidence interval. NPV: Negative predictive value. PPV: Positive predictive value. *: Statistically significant at *p* ≤ 0.05.

## Data Availability

Not applicable.

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
