# Peer review of "Are Antisense Long Non-Coding RNA Related to COVID-19?"

_biomedicines, 2022, doi:10.3390/biomedicines10112770_

Round 1

Reviewer 1 Report

A statistical  analysis  is needed to be sure there is not a multicomparison problem.  e paper has typographical  errors. There are several in the paper.

Author Response

Reviewer 1

  1. Reviewer comment: A statistical analysis is needed to be sure there is not a multi-comparison problem.

Response:

Kruskal Wallis, Kolmogorov–Smirnov and Shapiro-Wilk tests were used to verify the normality of distribution of variables. In addition, Post Hoc test (Tukey) was used for pairwise comparisons after ANOVA test for normally distributed quantitative variables and Post Hoc (Dunn's multiple comparisons test) for pairwise comparisons.

  1. Reviewer comment: Paper has typographical errors. There are several in the paper.

Response: The whole manuscript was carefully revised, and the grammatical and typographical errors were corrected and highlighted in yellow color for your revision and easy tracking. We truly appreciate the reviewer’s comments and We hope that the language is now acceptable for the next review process.

Reviewer 2 Report

In this manuscript Badr et al studied whether lncRNA has any role in COVID 19. Authors studied expression of some lncRNA with respect to COVID 19 severity by using RT PCR expression. The authors used ABCD score to as the measure of severity of the COVID 19 patients. I am having some concern about this manuscript which I have stated bellow.

1.     Please re-write the introduction. It’s very hard to follow.  Please avoid using comments li “but no one has investigated their expression based on COVID-19 severity”. There are lots of literature available between COVID severity and lncRNA. You can write with respect to ABCD score, no lncRNA expression data is available.

2.     Many of the reference is wrongly cited. The ABCD score reference is not correct one.

3.     Author need to explain scientifically why they selected the named lnRNAs for expression study.

4.     In RT-PCR studies for normalization, its better to use two endogenous housekeeping genes. From our experience using only one gene sometime give bias results.

5.     Authors need to discuss how the changes in expression has any role in the sevedrity of COVID 19.

Author Response

Reviewer 2

  1. Reviewer comment: Please re-write the introduction. It’s very hard to follow. Please avoid using comments li “but no one has investigated their expression based on COVID-19 severity”. There are lots of literature available between COVID severity and lncRNA. You can write with respect to ABCD score, no lncRNA expression data is available.

Response:

  • The introduction section was reconstructed and supported with more information about symptoms of COVID-19 and the previous studies concerned with the alteration of lncRNAs and the severity of COVID-19. In addition, the reviewer comment was taken into consideration.
  • Deletion of (“but no one has investigated their expression based on COVID-19 severity”)

Response: Done.

  1. Reviewer comment: Many of the references are wrongly cited. The ABCD score reference is not the correct one.

Response: All references were revised and arranged correctly and become relevant including that of ABCD score.

  1. Reviewer comment: The author needs to explain scientifically why they selected the named lnRNAs for the expression study.

Response:

The discussion section was amended and supported with more information that explain the relationship between the long noncoding RNA in this study and factors that affect viral covid infection like ferroptosis, free heme and iron and hypoxia induced factors.

  1. Reviewer comment: In RT-PCR studies for normalization, it is better to use two endogenous housekeeping genes. From our experience using only one gene sometimes give biased results.

Author Response: We agree that it is better to use two housekeeping genes. However, we only used the most common one (beta-actin) in this study and unfortunately there is no enough samples to perform more experiments.

  1. Reviewer comment: Authors need to discuss how the changes in expression have any role in the severity of COVID-19.

Response: The supplementary data were provided and presented in tables 2, s3 and s4 that showed the logistic regression analysis affecting mortality, mechanical ventilation, and complication. Also, the discussion section was supported with more information that discussed the significance of multivariate regression analysis of two of the studied long noncoding RNA with complication and artificial ventilation as an indicator for severity of the disease.

Reviewer 3 Report

The authors aimed at identifying the ex-pression level of antisense lncRNAs (A2M-AS1, DBH-AS1, FLVCR1-DT, and NCBP2AS2-1) and FLVCR1 in COVID-19 patients and its relation to severity of the disease.

- This study is very interesting.

- Introduction is well written, but it needs to be better focused on current and pertinent literature. Please expand this section.

- Methods section is well elaborated and clear.

- Results are well presented and particularly relevant.

- Discussion section is too superficial and should be improved and expanded commenting and citing more relevant articles of the current literature.

Author Response

Reviewer 3

  1. Reviewer comment: This study is very interesting. - Introduction is well written, but it needs to be better focused on current and pertinent literature. Please expand this section. - Methods section is well-elaborated and clear. - Results are well presented and particularly relevant.

 Response:

  • The introduction was reconstructed and supported with more information about symptoms of COVID-19 and the previous studies concerned with the alteration of lncRNAs and the severity of COVID-19.

  1. Reviewer comment: The discussion section is too superficial and should be improved and expanded by commenting and citing more relevant articles of the current literature.

Response:

The discussion section was amended and supported with more relevant information about lncRNA A2M antisense RNA1 (A2M-AS1, the Feline Leukemia, Subgroup C and Nuclear cap-binding protein subunit-2 antisense RNA 2 (NCBP2AS2) also called the hypoxia-induced angiogenesis regulator (HIAR) with different references.

Round 2

Reviewer 3 Report

amended manuscritp is acceptable.